# Effectiveness of Prompt Optimization in NL2SQL Systems

Sairam Gurajada*
sairam@megagon.ai
Megagon Labs
Mountain View, California, USA

Eser Kandogan
eser@megagon.ai
Megagon Labs
Mountain View, California, USA

Sajjadur Rahman*
sajjadurr@adobe.com
Adobe
San Jose, California, USA

## Abstract

NL2SQL approaches have greatly benefited from the impressive capabilities of large language models (LLMs). In particular, bootstrapping an NL2SQL system for a specific domain can be as simple as instructing an LLM with sufficient contextual information, such as schema details and translation demonstrations. However, building an accurate system still requires the rigorous task of selecting the right context for each query—including identifying relevant schema elements, cell values, and suitable exemplars that help the LLM understand domain-specific nuances. Retrieval-based methods have become the go-to approach for identifying such context. While effective, these methods introduce additional inference-time costs due to the retrieval process.

In this paper, we argue that production scenarios demand high-precision, high-performance NL2SQL systems, rather than simply high-quality SQL generation, which is the focus of most current NL2SQL approaches. In such scenarios, the careful selection of a static set of exemplars—capturing the intricacies of the query log, target database, SQL constructs, and execution latencies—plays a more crucial role than exemplar selection based solely on similarity. The key challenge, however, lies in identifying a representative set of exemplars for a given production setting. To this end, we propose a prompt optimization framework that not only addresses the high-precision requirement but also optimizes the performance of the generated SQL through multi-objective optimization. Preliminary empirical analysis demonstrates the effectiveness of the proposed framework.

## ACM Reference Format:

Sairam Gurajada, Eser Kandogan, and Sajjadur Rahman. 2025. Effectiveness of Prompt Optimization in NL2SQL Systems. In *Novel Optimizations for Visionary AI Systems (NOVAS '25), June 22–27, 2025, Berlin, Germany*. ACM, New York, NY, USA, 6 pages. https://doi.org/10.1145/3735079.3735325

## 1 Introduction

The rapid progress in LLM capabilities—specifically their ability to follow instructions and maintain large contexts—has made them a natural choice in many applications. Natural Language to SQL (NL2SQL) is a long-standing and important task in many business-critical scenarios, requiring a deep understanding of user queries

---

*Work done while at Megagon Labs

and the underlying databases for effective translation. Recent years have witnessed significant progress in NL2SQL, fueled by advancements in LLMs [3, 4, 8, 15]. However, building an effective NL2SQL system goes beyond simply leveraging LLMs—it requires the careful selection of instructions, exemplars, and schema, making it a challenging task despite recent breakthroughs [6].

Recent works [4, 21] emphasize that exemplar selection is crucial for building effective NL2SQL systems. Retrieval-based exemplar selection—i.e., identifying exemplars similar to the user query—has become the de facto method. However, studies [4, 19] highlight inefficiencies and overfitting issues with similarity-based retrieval methods, and argue that synthetic exemplars can yield better performance. While each approach has its advantages—retrieval-based methods are cheaper due to index-based lookups without LLM calls, and synthetic exemplars may be more accurate—they both require exemplar selection at inference time, which can become a bottleneck in business-critical applications [23].

**Prompt Optimization.** To address the limitations of current NL2SQL systems, we argue that for effective SQL generation, all an LLM needs is a static set of exemplars that capture the intricacies of the domain—offering performance comparable to retrieval-based approaches, while eliminating the need for inference-time retrieval. The key challenge lies in identifying this representative set of exemplars. To tackle this, we leverage prompt optimization techniques for exemplar selection in NL2SQL and demonstrate their effectiveness.

**Multi-Objective Optimization.** Most existing NL2SQL approaches focus solely on accuracy. However, accuracy is only one dimension in deploying practical NL2SQL systems. In real-world settings, systems must also understand query efficiency and the characteristics of target SQL engines, generating queries that are efficient to execute (i.e., with lower latency). In this work, we propose a way to extend prompt optimization to multi-objective settings. To support this, we introduce an augmented benchmark based on BIRD that includes query latency measurements.

To summarize, our contributions are as follows:

- To the best of our knowledge, this is the first work to study the effectiveness of prompt optimization in NL2SQL systems.
- We propose an iterative prompt optimization (IPO) framework that jointly optimizes instructions and exemplar selection through two agents *Proposer* and *SQL Generator*. Additionally, the framework implicitly performs schema pruning, reducing prompt size and thereby lowering inference costs.
- We introduce the aspect of generating efficient SQL translations in NL2SQL systems, and introduce an augmented benchmark BIRD-MULTI (based on BIRD dataset) that incorporates query latency information.

## 2 Related Work

**Exemplar Selection.** With the advent of powerful API-based LLMs such as ChatGPT [25] and Gemini [24], in-context learning (ICL)–based approaches [4, 5, 7, 16, 19–21] have become the dominant strategy for building high-performing NL2SQL systems. Specifically, retrieval-based exemplar selection [21], where examples are selected from a training set based on text or structural similarity, has proven sufficient to improve NL2SQL performance without expensive fine-tuning. However, such systems introduce inference-time costs and may overfit to specific queries due to the retrieval of overly similar examples [4].

To address this, recent approaches [4, 19] employ (online) synthetic exemplar generation rather than relying on training data selection. While this mitigates overfitting, it requires learning exemplar generators, which incurs additional costs and presents challenges in domain transfer. In this work, we explore optimization-based methods for exemplar selection that avoid both the expense of retrieval indexes and the complexity of online synthetic generation.

**Prompt Optimization.** Optimizing LLM prompts has been a focus for several years [22, 26], showing effectiveness across a multitude of applications. More recently, DSPy [9] introduced a declarative framework for expressing and optimizing prompts for NLP tasks. Foundational work by [27] demonstrated the inherent capability of LLMs to act as optimizers, particularly for instruction tuning across various tasks. Building on this,[18] proposed MIPRO, a non-iterative technique for joint optimization of instructions and exemplar selection in multi-stage pipelines. Furthermore,[13] introduced a declarative framework focused on BI workloads, combining hybrid database systems with AutoML-style optimization for pipeline tuning. While these works introduced key optimization techniques, their applicability and effectiveness in the NL2SQL setting remain unexplored—a gap that our work seeks to address.

## 3 Prompt Optimization for NL2SQL

To demonstrate the effectiveness of optimization in NL2SQL, we adopt a simple in-context learning (ICL)[2] pipeline, as illustrated in Figure1, which uses a single LLM to generate the SQL query. The prompt provided to the LLM consists of: a) #Instruction – a guiding instruction for the task, b) #Exemplars – examples selected from the training data via an Exemplar Selection component, c) #Query – the user query to be translated, d) #Schema – the relevant schema retrieved using a Schema Retrieval module, and e)#SQL – a prefix to trigger SQL generation by the LLM.

To apply to production use-case, we use exact proprietary schema, and focus our efforts on optimizing exemplar selection.

### 3.1 Exemplar Selection and Optimization

As previously mentioned, exemplar selection is a crucial step in NL2SQL generation, particularly when using ICL-based approaches [8]. This involves identifying an appropriate set of exemplars—each consisting of a natural language (NL) query, database schema, corresponding SQL query, and optionally hints or cell values—that help the LLM understand the domain, the target SQL engine, and data-specific nuances. Below, we discuss various exemplar selection strategies and how optimization can enhance the selection process.

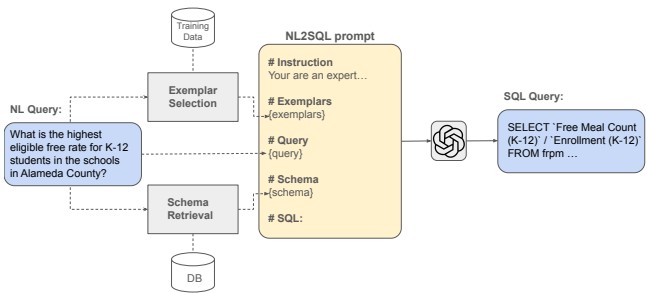

**Figure 1: NL2SQL Pipeline**

---

**Algorithm 1:** Optimization of Random Exemplar Selection

**Input:** $\mathcal{D}$, $numTrials$, $\mathcal{P}_{base}$, LLM
**Output:** $\mathcal{P}_{opt}$

1 $\mathcal{D}_{train}, \mathcal{D}_{valid} \leftarrow \texttt{Split}(\mathcal{D})$
2 $best\_score \leftarrow 0$
3 $P_{opt} \leftarrow \mathcal{P}_{base}$
4 **while** $i \leq numTrials$ **do**
5    $k \leftarrow trial.\texttt{suggest\_int}('k', 0, K)$ /* choose $k$     */
6    $E_i \leftarrow \texttt{Random}(\mathcal{D}_{train}, k)$ /* $k$ exemplars     */
7    $\mathcal{P}_i \leftarrow \mathcal{P}_{base} + Ex_i$
8    $score \leftarrow \texttt{Evaluate}(\text{LLM}, \mathcal{P}_i, \mathcal{D}_{valid})$
9    **if** $score > best\_score$ **then**
10       $P_{opt} \leftarrow \mathcal{P}_i$
11       $best\_score \leftarrow score$

---

**Random.** A straightforward approach to exemplar selection is random sampling. For a predefined value of $k$ (the number of exemplars), this strategy randomly samples $k$ exemplars from the training data to include in the prompt. More sophisticated sampling techniques, such as stratified sampling, can also be used to account for the distribution of query types. For example, queries in the BIRD [11] dataset are categorized into three groups: simple, moderate, and challenging.

**Optimizing Random Exemplar Selection.** A key challenge with random selection is choosing an appropriate value for $k$. A small $k$ may fail to capture the diversity of the NL and SQL constructs, while a large $k$ can lead to lost-in-the-middle issues with LLMs [14] and increase generation costs due to the larger prompt size. A simple yet effective approach is to treat $k$ as a hyperparameter and optimize it using AutoML-style techniques, as illustrated in Algorithm 1. Inspired by DSPy's BootStrap with FewShot Example Selection [9], this method optimizes the number of demonstrations by randomly sampling exemplars (with replacement), rather than bootstrapping, using a performance metric $\mu$.

In addition to optimizing exemplar selection, joint optimization of instruction and exemplar selection can lead to improved performance. MIPRO [18] leverages an LLM to generate $N$ instruction–exemplar pairs $(I_1, E_1), (I_2, E_2), \ldots, (I_N, E_N)$, where $I_i$ is an instruction generated from a set of randomly bootstrapped exemplars $E_i$. A hyperparameter optimization algorithm such as TPE [1]

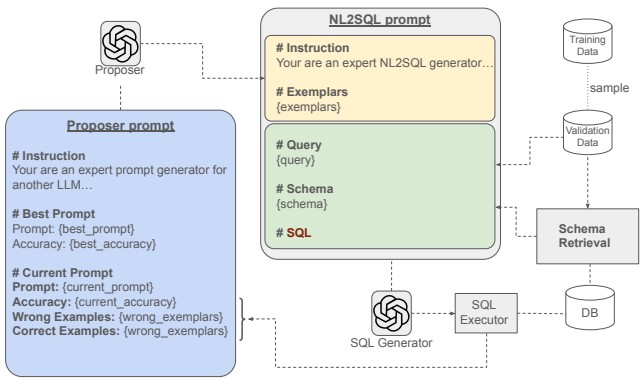

**Figure 2: Iterative Prompt Optimization**

is then used to identify the optimal pair $(I_i, E_i)$ based on an objective function, such as validation accuracy.

## 3.2 Iterative Prompt Optimization

One of the key limitations of the exemplar selection strategies discussed earlier is their ad hoc nature—exemplars are either randomly sampled or bootstrapped using heuristics (as in [18]), which may lead to suboptimal performance. Long-context LLMs (LCMs) aim to overcome this by fitting a larger number of exemplars (100–200) into their context window. However, recent work [4] has shown that relying on LCMs to implicitly perform exemplar selection does not improve performance and can, in fact, be detrimental, as LCMs often struggle with effective in-context learning [12].

To address this, we extend the work of [27], which uses an LLM as an optimizer to find an optimal prompt instruction, by enabling it to perform both instruction generation and exemplar selection through two cooperating agents: the Proposer and the SQL Generator. Specifically, we introduce an Iterative Prompt Optimization (IPO) approach (illustrated in Figure 2) in which the two agents work together to discover optimal NL2SQL prompts for a given training corpus.

The Proposer agent takes a Proposer prompt as input and generates an NL2SQL prompt comprising an instruction and a set of exemplars. The SQL Generator agent then evaluates the generated prompt on a validation set (sampled iteratively from the training data) and collects performance metrics including accuracy, as well as the correct and incorrect examples. This feedback is used to update the Proposer prompt. In subsequent iterations, the Proposer is guided to refine the NL2SQL prompt based on past performance, aiming to produce more informative exemplars and a better-suited instruction for improved SQL generation.

In contrast to MIPRO, which bootstraps exemplars randomly, IPO uses an LLM as an optimizer to jointly refine both the instruction and exemplar selection. Additionally, we observed that IPO often generates more concise NL2SQL prompts by pruning irrelevant schema information from the exemplars. For example, Figure 3 shows an exemplar whose schema includes only the table `film` and the columns `film_id`, `title`, and `rating` from the database `movie_3`. Although schema pruning was not an explicit design goal of IPO, this behavior highlights the strength of LLMs as optimizers in complex tasks such as NL2SQL.

```
NLQ: List all the films that are rated as PG-13.
Schema:
Database Name: movie_3
Tables: ['film']
#Columns:
film: [film_id:integer, title:text, rating:text]
Evidence: film refers to title; rated as PG-13 refers to
rating = `PG-13`.
SQL: SELECT title FROM film WHERE rating = 'PG-13';
```

**Figure 3: IPO generated exemplar with automatic schema pruning**

## 4 Extending to Multi-Objective

**Motivation.** Thus far, NL2SQL systems have focused mainly on improving execution accuracy while ignoring a critical dimension: generating efficient SQL queries. Consider the example of SQL translations: ground truth (GT) and generated SQL (Gen) for the query NLQ. Executing the GT SQL query on a SQLite3 database took around 10.2 seconds (due to the sub-query), while the Gen query (which uses an inner join) took only 0.03 seconds. This example demonstrates that the generated query (in this case, ground truth) may not always be the most efficient SQL translation.

```
NLQ: Show the avatar of the user who gave the rating at
2019/10/17 1:36:36.
GT: SELECT user_avatar_image_url FROM lists_users
    WHERE user_id = (SELECT user_id FROM ratings
    WHERE rating_timestamp_utc LIKE '2019-10-17 01:36:36')
Gen:SELECT T2.user_avatar_image_url
    FROM ratings AS T1 INNER JOIN lists_users AS T2
    ON T1.user_id = T2.user_id
    WHERE T1.rating_timestamp_utc LIKE '2019-10-17 01:36:36'
```

**Benchmark Creation.** To build NL2SQL systems capable of generating efficient SQL queries, it is essential to have information about the efficiency of a SQL query—such as its wall-clock execution time—alongside the SQL translation itself. This efficiency information enables SQL generators (LLMs) to better understand the nuances and computational complexity of various SQL constructs, ultimately guiding them toward generating more optimized queries. However, existing benchmarks such as BIRD[11] and SPIDER[10] lack this critical execution-time data. To address this gap, we developed a new augmented benchmark built on top of BIRD, which includes each natural language query (NLQ) paired with two different SQL variants (generated using reasoning models OpenAI O3 [17]) along with their measured execution times.

```
NLQ: Give the full name of the actor with the highest rental rate.
SQL1: SELECT a.first_name, a.last_name FROM actor AS a JOIN
film_actor AS fa ON a.actor_id = fa.actor_id JOIN film AS f ON
fa.film_id = f.film_id ORDER BY f.rental_rate DESC LIMIT 1;
Time1: 0.0012 seconds
SQL2: SELECT a.first_name, a.last_name FROM actor a JOIN
film_actor fa ON a.actor_id = fa.actor_id JOIN film f ON
fa.film_id = f.film_id WHERE f.rental_rate =
(SELECT MAX(rental_rate) FROM film) LIMIT 1;
Time2: 0.0009 seconds
```

**Generating Efficient SQL Queries.** With the augmented benchmark containing SQL variants and their corresponding execution times, it becomes feasible to design LLM prompts specifically aimed at generating efficient SQL queries. Furthermore, by leveraging the optimization techniques described in Section 3, it is possible to jointly optimize for both SQL efficiency and generation accuracy, leading to more practical and performant NL2SQL systems.

## 5 Preliminary Results

Here, we present preliminary results demonstrating the effectiveness of prompt optimization in NL2SQL systems. As described earlier, we consider a simple NL2SQL pipeline consisting of a single LLM (illustrated in Figure 1) and evaluate the following prompt optimization strategies discussed in Section 3. For all experiments, we use GPT-4o as the LLM.

- **RES.** Random Exemplar Selection (RES) is the baseline approach, where $k = 10$ exemplars are randomly sampled from the training data. We run RES over 10 random samples and report the best accuracy.
- **ORES.** Optimized Random Exemplar Selection (ORES) uses Bayesian Optimization to tune the hyperparameter $k$ in the RES method. We limit the number of trials to 20 and note that increasing trials does not necessarily correlate with improved performance.
- **MIPROv2.** Multiprompt Instruction PRoposal Optimizer (MIPROv2) is the DSPy [9] recommended optimizer that jointly optimizes instruction and exemplar selection. Similar to ORES, we set the number of trials to 20, and the maximum number of labeled demonstrations to 10.
- **IPO.** Iterative Prompt Optimization (IPO) is a bi-agent, LLM-as-optimizer approach that iteratively refines the NL2SQL prompt using feedback on SQL generation quality. For IPO, we set the number of iterations to 5 and instruct the LLM to generate at least 5 diverse exemplars per iteration.

### 5.1 Effectiveness of Optimization

**Performance.** Table 1 highlights the effectiveness of different optimization strategies on the BIRD dataset. The naive RES approach underperforms due to its inability to select informative exemplars for SQL generation. The ORES approach, which applies a simple AutoML-based optimization, performs better than RES but falls short compared to more advanced strategies like MIPROv2 and IPO. IPO achieves the best performance, benefiting from iterative refinement using feedback from the SQL Generator agent, which leads to more relevant exemplar selection. The absence of this feedback loop in MIPROv2 makes it less effective than IPO. However, we emphasize that these observations are specific to the NL2SQL task.

**Quantitative Analysis.** Table 2 presents a quantitative analysis of the different optimization techniques, measured across two dimensions: prompt length and optimization time. RES (with 10 exemplars) results in a prompt length of approximately 23k tokens, while ORES (with 75 exemplars) leads to a significantly larger prompt of around 84k tokens. MIPROv2 (with 10 exemplars) produces a prompt length of about 26k tokens, similar to RES. IPO yields the shortest prompt length, as it prunes a substantial portion of schema information from each exemplar, while also delivering the best performance.

| Approach | Simple | Moderate | Challenging | Total |
|---|---|---|---|---|
| RES | 52.32 | 48.17 | 43.35 | 50.74 |
| ORES | 58.27 | 51.29 | 46.21 | 55.02 |
| MIPROv2 | **64.86** | 48.28 | 44.14 | 57.89 |
| IPO | 63.57 | **52.28** | **45.52** | **59.24** |

**Table 1: Ex. Accuracies of Prompt optimization Methods on BIRD (dev) dataset**

In terms of optimization time, MIPROv2 takes the longest, as it involves both data analysis and joint optimization, whereas IPO and ORES are comparatively faster.

| Approach | Prompt Length (#tokens) | Optimization Time |
|---|---|---|
| RES | 23,749 | - |
| ORES | 84,519 | **5m26s** |
| MIPROv2 | 26,352 | 13m16s |
| IPO | **6,495** | 8m53s |

**Table 2: Quantitative analysis of PO on BIRD (dev) dataset**

### 5.2 Multi-Objective Optimization

Table 3 demonstrates the effectiveness of multi-objective optimization using the IPO approach. For this, we consider both accuracy and latency on the BIRD (dev) dataset. When compared to the ground truth (GT), accuracy-only IPO optimization (Section 3.2) results in the generation of SQL queries that are less efficient, with a maximum latency of approximately 18 seconds (vs. 8.7 seconds for GT) and a standard deviation ($\sigma$) that is almost 1.8 times larger. In contrast, joint optimization of both accuracy and latency leads to only a marginal increase in the maximum latency of queries, while maintaining a standard deviation similar to that observed in GT.

| Approach | Execution Time(s) | | | | Acc. | Gen. Time |
|---|---|---|---|---|---|---|
| | Min | Max | $\sigma$ | $\mu$ | (Ex) | QPS |
| GT | 1e-5 | 8.761 | 0.352 | 0.0026 | - | - |
| Acc.Opt | 9e-5 | 18.512 | 0.635 | 0.0051 | 59.24 | 1.56 |
| + Lat. | 7e-5 | 10.156 | 0.383 | 0.0028 | 58.98 | 1.72 |

**Table 3: Effectiveness of multi-objective optimization on BIRD (dev) dataset**

## 6 Conclusions

In this paper, we propose a novel approach for building NL2SQL systems by leveraging prompt optimization as a key strategy. Specifically, we optimize both instruction and exemplar selection for LLM-based SQL generation. Furthermore, through prompt optimization, we demonstrate that schema pruning and concise prompt generation can be achieved without negatively impacting accuracy. Additionally, we highlight the importance of generating efficient SQL queries and show that prompt optimization can effectively achieve the dual objectives of accuracy and efficiency. Finally, we introduce an augmented benchmark designed for the multi-objective optimization of NL2SQL systems.

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

# A Appendix

## A.1 Prompt Templates

- **Random and Optimizes Exemplar Search (RES & ORES)**

```
#Instruction
Given natural language query, schema of the database
and evidence, generate a sqlite SQL query

#Exemplars
NLQ: {{NLQ}}
SCHEMA: {{DB_SCHEMA}}
EVIDENCE: {{EVIDENCE}}
SQL: {{SQL}}

#Query
NLQ: {{NLQ}}
SCHEMA: {{DB_SCHEMA}}
EVIDENCE: {{EVIDENCE}}
```

- **Proposer Agent**

```
#Instruction
You are an expert in assisting another LLM for the
task of generating SQL queries from natural language
queries. You are given the following information:
1. The best prompt generated so far
2. The accuracy of the best prompt
3. The current prompt
4. The accuracy of the current prompt
5. A set of exemplars where the current prompt
incorrectly generated the SQL query
6. A set of exemplars where the current prompt
correctly generated the SQL query

#Goal:
Think step by step to generate a prompt comprising
of two parts in JSON format:
1. Instruction for the LLM to generate SQL query
for sqlite3 database
2. A set of diverse exemplars to assist the LLM in
generating the SQL query.

#Best Prompt:
{best_prompt}

#Best Accuracy:
{best_accuracy}

#Current Prompt:
{current_prompt}

#Current Prompt Accuracy:
{current_accuracy}

#Wrong Exemplars:
{wrong_examples}

# Correct Exemplars:
{correct_examples}

# Proposed Prompt:
```

- **Proposed Prompt: Example**

```
#Instruction
Given a natural language query (NLQ), the schema of
the database, and relevant evidence, generate a
valid SQLite SQL query that satisfies the NLQ. Use
the provided schema and evidence to ensure the SQL
query correctly answers the NLQ. Only utilize
relevant columns and tables in the query.
Return only the SQL query without any prefixes
or block quotes.

#Exemplars:
NLQ: List all the films that are rated as PG-13.
Schema:
Database Name: movie_3
Tables: ['film']
#Columns:
film: [film_id:integer, title:text, rating:text]
Evidence: film refers to title; rated as PG-13 refers
to rating = 'PG-13'.
SQL: SELECT title FROM film WHERE rating = 'PG-13';

...
```

- **Variant Generator for Multi-Objective Benchmark**

```
#Instruction
Given natural query, database schema, corresponding SQL,
generate {num_variants} SQL variants.
Generate only valid SQL query without any prefix
or suffix:

#Query
{query}

#Database Schema
{db_schema}

#Ground Truth SQL:
{sql}

#SQL Variants:

1.

2.

3.
```