# OpenReview forum: "Effectiveness of Prompt Optimization in NL2SQL Systems"
_ACM.org/SIGMOD/2025/Workshop/NOVAS — NOVAS 2025_

### Official Review · Reviewer_VEn6 · 2025-04-20

**Confidence:** 5

**Improvement Opportunities:**

Clarify benchmark novelty: While Section 4 introduces BIRD-MULTI, it appears to be a latency-augmented version of BIRD. It is unclear how the benchmark was constructed---did the authors generate alternative SQL variants or just time existing ones? And, how did the authors come up with the alternative SQL variants?

Cost of optimization unclear: IPO uses 5 iterations, whereas MIPRO runs for 20 trials. Why is IPO faster to converge? Does it perform fewer LLM calls per trial? Clarifying the optimization cost tradeoffs (e.g., total tokens generated or $ cost) would be valuable.

Experimental completeness: The evaluation omits a relevant DSPy baseline, such as COPRO, which is known for multi-objective optimization. While it's understandable that space is limited, the paper should acknowledge why COPRO was excluded or deemed not comparable.

**Minor Comments:**

Fig 3: "exampler" should be "exemplar?"

Section 5.2 contains a dangling citation "(Section ??)"

**Short Summary:**

This paper explores the effectiveness of prompt optimization for building NL2SQL systems. Instead of relying on retrieval-based exemplar selection at inference time, which would incur additional latency, the authors propose selecting a static set of instructions and few-shot examples, with an optimization approach that jointly optimizes exemplar selection and instructions (akin to DSPy). The authors develop a method called Iterative Prompt Optimization (IPO), which employs two AI agents to improve both the instructions and examples used in prompts. The paper also augments the BIRD benchmark with additional logically equivalent queries and their query execution times, to be used in finding efficient few-shot examples.

**Strong Points:**

Timely motivation: The paper makes a strong case that real-world NL2SQL systems must prioritize execution efficiency in addition to accuracy (Section 4). The motivating example where an alternative, logically-equivalent SQL query is 340x faster than the GT query is effective and clear.

Clear explanation of prompt optimization: Section 3 provides a well-articulated breakdown of prompt components and optimization strategy.

Concise and relevant prompts: IPO-generated prompts are shown to be much shorter (6.5k tokens vs 23k–84k). Many LLM-generated prompts tend to be verbose.

---

### Official Review · Reviewer_QCdw · 2025-04-21

**Confidence:** 4

**Improvement Opportunities:**

I1: The paper would benefit from a comparison with RAG systems, both in terms of latency and accuracy. This is needed to support the motivation in the Introduction of the paper on why prompt optimization is needed over RAG.

I2: The paper only evaluates on the BIRD dataset. How does this compare on more complex benchmarks like Beaver, Spider 2.0? This would help the reader evaluate the approach better.

I3: Clarification question on Table 3: Is the sigma and mu measured across multiple runs of the entire benchmark, or across queries? Why is the sigma so high? Can you also report the latency numbers of other baselines?

**Minor Comments:**

Alg1: numTrails -> numTrials; trail -> trial

Sec 5.2 - There is an invalid reference that is rendered as ??

"Your are" -> "You are" in the instruction prompt in all figures.

**Short Summary:**

This paper proposes prompt optimization as a novel solution for text-to-sql. Unlike RAG-style approaches, it uses iterative prompt optimization to select exemplars to include in the prompt. The method supports multi-objective optimization, targeting both accuracy and execution latency. Experiments show promising improvements over existing baselines. Overall, the paper studies a relevant problem, presents an interesting solution, and provides encouraging preliminary results. I recommend Accept.

**Strong Points:**

S1: The paper presents a novel framework for text-to-sql that uses prompt optimization. The proposed iterative prompt optimization solution is intuitive, and empirical results are promising.

S2: The paper focuses on both accuracy and query latency. This multi-objective perspective is relevant and practical.

---

### Official Review · Reviewer_Hyiq · 2025-04-27

**Confidence:** 4

**Improvement Opportunities:**

1. Limited Evaluation on only one benchmark
2. Missing Comparison to Stronger Baselines
3. Insufficient Algorithmic Detail for IPO
4. Lack of More Detailed Analysis

**Minor Comments:**

D1. The experiments are restricted to the BIRD dataset and the new BIRD-MULTI variant. While insightful, broader evaluation on more diverse datasets (e.g., SPIDER 1.0 and SPIDER 2.0) is missing, which limits the generalizability of the findings.

D2. Although the paper compares with RES, ORES, and MIPROv2, it omits more recent and competitive retrieval-augmented generation or NL2SQL methods.

D3. While the general pipeline of IPO is explained (Figure 2), the detailed functioning of the Proposer agent, especially how it uses feedback to revise prompts based on correct/wrong examples, is not clear.

D4. The iterative optimization process of IPO is interesting, but the paper does not provide convergence curves (e.g., validation accuracy per iteration) or discussions about optimization stability (e.g., risks of local minima or performance oscillations).

**Short Summary:**

This paper studies the problem of Natural Language to SQL (NL2SQL) by proposing a prompt optimization framework. To this end, the authors introduce Iterative Prompt Optimization (IPO), an LLM-driven method that iteratively refines both instructions and exemplars to optimize SQL generation. Beyond simply maximizing execution accuracy, the paper extends prompt optimization to a multi-objective setting, considering SQL execution latency. To evaluate the methods, the paper develops a benchmark dataset BIRD-MULTI for evaluating both query correctness and execution efficiency. Preliminary experiments demonstrate that IPO outperforms existing approaches such as RES, ORES, and MIPROv2 in both accuracy and efficiency while significantly reducing prompt length.

**Strong Points:**

1. Clear Problem Motivation and Practical Relevance
2. Novel and Practical Optimization Framework (IPO)
3. Extension to Multi-Objective Optimization